# Tele-Exercise for Fitness: Physical and Psychological Outcomes in Athletes and Non-Athletes’ Trainees

**DOI:** 10.3390/healthcare13040354

**Published:** 2025-02-07

**Authors:** Enzo Iuliano, Giovanna Zimatore, Antonio Fabbrizio, Andrea De Giorgio, Martina Sausa, Barbara Maria Matteo, Sonia Angilletta, Victor Machado Reis, Nuno D. Garrido, José Vilaça-Alves, Rafael Peixoto, Paulina Perusina, Aleksandra Aristova, Carlo Baldari, Filippo Macaluso, Alberto Fucarino, Manuela Cantoia

**Affiliations:** 1Department of Theoretical and Applied Sciences, eCampus University, 22060 Novedrate, Italy; enzo.iuliano@uniecampus.it (E.I.); giovanna.zimatore@uniecampus.it (G.Z.); antonio.fabbrizio@uniecampus.it (A.F.); andrea.degiorgio@uniecampus.it (A.D.G.); martina.sausa@uniecampus.it (M.S.); barbara.matteo@uniecampus.it (B.M.M.); sonia.angiletta@uniecampus.it (S.A.); carlo.baldari@uniecampus.it (C.B.); alberto.fucarino@uniecampus.it (A.F.); manuela.cantoia@uniecampus.it (M.C.); 2Research Center in Sports Sciences, Health Sciences and Human Development (CIDESD), 5001-801 Vila Real, Portugal; vmreis@utad.pt (V.M.R.); ngarrido@utad.pt (N.D.G.); josevilaca@utad.pt (J.V.-A.); peixoto347@gmail.com (R.P.); 3Sport Sciences Departments, University of Tra’s-os-Montes e Alto Douro, 5000-801 Vila Real, Portugal; 4Rijeka Sports Association for Persons with Disabilities, 51 000 Rijeka, Croatia; paulina.perusina@ssoi-rijeka.hr; 5Latvian Sports Federations, LV-1013 Riga, Latvia; aleksandra.aristova@lsfp.lv; 6Department of Biomedicine, Neurosciences and Advanced Diagnostics, University of Palermo, 90133 Palermo, Italy

**Keywords:** tele-exercise, physical fitness, well-being, disabilities, lifestyle, training monitoring, stress, remote training

## Abstract

**Background**: This research explored how tele-exercise influenced physical fitness and mental well-being in individuals with and without disabilities and with different training habits. **Methods**: One hundred and ninety-three participants were categorized into two groups: athletes and non-athletes. Participants were involved in either synchronous or asynchronous tele-exercise programs of two or three sessions of workouts per week, lasting eight weeks. Psychological well-being was evaluated pre- vs. post-intervention by the WHO-Five Well-Being Index (WHO-5), Psychological General Well-Being Index (PGWBI), and Perceived Stress Scale (PSS-10). Fitness assessment tools included the 2-minute step test, curl-up test, forward bending test, and squat test. Body weight was also assessed pre- and post-intervention. **Results**: The analysis showed a significant increment in all fitness scores (*p*-value ranged from <0.001 to 0.015) with no change in body weight. Psychological tests indicated an overall increment in the participants’ well-being measured by the WHO-5 and unexpectedly, also in the stress level assessed through PSS-10. **Conclusions**: Enabled by technology, tele-exercise provides a flexible and convenient complementary approach to conventional workouts, helping individuals maintain an active lifestyle and improve their well-being. Positive effects on the sedentary population’s lifestyles are discussed.

## 1. Introduction

Over the past few decades, the world of physical fitness and exercise has undergone a remarkable transformation with the advent of tele-exercise. The COVID-19 pandemic served as a catalyst, hastening the acceptance and integration of tele-exercise into mainstream fitness culture. Lockdowns and social distancing measures prompted individuals to seek alternative methods to maintain their physical activity levels, resulting in a substantial surge in demand for online fitness programs and virtual classes [1]. During this complex time period, the use of tele-exercise brought not only physical benefits, but also considerable psychological support [2,3,4]. However, it would be profoundly incorrect to claim that tele-exercise emerged at the end of 2019. Years earlier, research conducted by Vandelanotte et al. [5] emphasized the effectiveness of online platforms in promoting physical activity and enhancing health outcomes. These platforms provide unparalleled flexibility and convenience, allowing individuals to participate in guided workouts from the comfort of their own homes. The incorporation of live streaming technology and fitness apps has accelerated the rise of tele-exercise, facilitating real-time interaction with trainers and fostering a sense of community among virtual fitness enthusiasts. This combination of immediacy and accessibility enhances the workout experience, making it more engaging and supportive [6]. If we briefly analyze the history behind the emergence of tele-exercise, we can see that it originated from telemedicine and later diverged from it [7,8]. From its early stages, tele-exercise demonstrated unique qualities that made it a valuable tool for professionals in the individual health field. Once this initial stage passed, there was an exponential increase in the utilization of tele-exercise as a supportive measure in treating various diseases. An increasing number of studies and trials have incorporated tele-exercise within their protocols across diverse subject categories [9,10,11,12,13]. However, at that time, the widespread adoption of tele-exercise within the general healthy population remained a distant prospect [14]. Tele-exercise has seen significant growth, largely driven by technological advancements.

Before the COVID-19 pandemic, the technology was already adequate, but there was a lack of interest from the general population. Initially, fitness communities responded to the sudden demand for tele-exercise by utilizing generic applications like Skype, Zoom, and Facetime as well as social networks such as TikTok, Instagram, and Facebook. Once tele-exercise had been cleared through customs within daily life, more and more ways to practice it began to emerge through specialized apps and online platforms [15]. As a result, the scientific community began to see tele-exercise as a viable mode of enjoyment that needed in-depth analysis, which had not yet been accomplished on healthy people until now. Post pandemic, specific research on healthy subjects has shown the benefits of tele-exercise both physically and psychologically [16,17,18]. However, to date, the number of studies that have been conducted is still small, which needs to increase to directly highlight the benefits resulting from tele-exercise activity. It is from this background that the TELEexe4ALL project was born. Our project’s main objective is to explore the physical and psychological effects of tele-exercise in a general and inclusive population while promoting the enjoyment of online physical activity through an ad hoc platform, with the aim of demonstrating the benefits of tele-exercise in promoting and teaching a comprehensive healthy lifestyle. The main authors’ hypotheses are that tele-exercise can have an impact on psychological well-being levels (HP1), and that it can be a feasible and valuable support to set a healthy physical condition in the general population (HP2).

Since tele-exercise lacks the direct supervision found in a gym or clinical setting, it is critical to monitor physiological indices to ensure both safety and effectiveness.

Three types of physical activity (PA) assessment methods can be distinguished: criterion methods, objective methods, and subjective methods [19,20]. In this work, we assessed PA during tele-exercise both in asynchronous and synchronous participation by subjective methods, as explained in the following paragraph.

## 2. Materials and Methods

### 2.1. Study Design

This study was an interventional pre-post-test quasi-experimental study with no control group.

### 2.2. Participant

A total of 193 participants of both genders volunteered for the present study, aged 36.73 ± 14.61 years old. These participants were classified into 2 groups: 126 non-athletes and 67 athletes. Each participant had the option to follow a tele-exercise program either asynchronously or synchronously. The main characteristics of the participants as well as the distribution of those who chose asynchronous or synchronous participation are presented in Table 1.

Four age groups were defined in order to test the Age factor (see Section 2.4):Young adult: 18–24 years old;Adult: 25–44 years old;Middle age: 45–59 years old;Elderly [>60] years old.

Enrollment was performed using a convenience sampling, and the participants were assigned to the two intervention groups (non-athletes and athletes) based on the inclusion and exclusion criteria outlined in Table 2.

Recruitment was promoted via the TELEexe4ALL open-source platform, the project’s social media channels (Twitter, Facebook, Instagram) as well as the social channels of the TELEexe4ALL partners to attract potential participants. Candidates completed a personal data questionnaire to assess their eligibility according to the inclusion and exclusion criteria. Based on their outcomes, they were allocated to one of the two intervention groups.

### 2.3. Procedures

The intervention lasted for 8 weeks, repeated several times during the year. During this period, the group of non-athlete participants attended 24 telematic exercise sessions (3 sessions per week on non-consecutive days), each lasting 1 h. During the same period, participants in the athlete group supplemented their regular training routine with 16 exercise sessions (2 sessions per week on non-consecutive days). All sessions were delivered via the TELEexe4ALL open-source platform. After registering, the participants could choose between synchronous or asynchronous modes and had access to the sessions scheduled for their respective groups.

*Synchronous mode:* Participants followed video-based exercises while being supervised by an experienced instructor who provided real-time feedback and corrections. This mode allowed the participants to interact with the instructor and others in real-time through the TELEexe4ALL platform’s meeting functionality. The instructors clearly demonstrated each exercise, explaining the proper technique and common mistakes to avoid. All exercises were designed to be performed at home without gym equipment or machinery. The instructor-to-participant ratio ranged from 1:2 to 1:8 during these sessions. This ratio varied depending on the scheduling of the lessons, which were conducted in multiple sessions. Therefore, the ratio was generally 1:8, but in some cases, due to lesson rescheduling by some participants, this ratio could occasionally drop to as low as 1:2. This choice was necessary to ensure that all participants could complete 100% of the scheduled lessons without absences.

*Asynchronous mode:* In this mode, the participants did not receive real-time supervision. Instead, they could access the same video lessons as the synchronous group at their own convenience and complete the exercises independently. The platform tracked their video access to ensure adherence to the schedule and prevent overtraining or undertraining. As with the synchronous sessions, all exercises were suitable for home settings without the need for equipment.

The training content was tailored to the specific needs of each group. The athlete group focused on strength and conditioning exercises aimed at injury prevention, targeting major muscle groups and joints. The workout intensity started at 4 RPE (rate of perceived exertion) on the Borg CR10 scale, progressing to a maximum of 6 RPE. The non-athlete group had the possibility to engage in different training programs depending on their needs.

All of the proposed training programs were designed to improve cardiovascular and muscular fitness, emphasizing the major muscle groups for maximum health benefits as well as coordination skills for overall motor skill development. However, the overall intensity of the workout was the same and ranged from 4 RPE to 8 RPE, depending on the fitness levels of the non-athlete participants.

#### 2.3.1. Subjective Assessments

At the start and end of the 8-week intervention, the participants underwent a series of assessments to measure the pre-/post-intervention outcomes with no restrictions on completion time (see Figure 1). These included both physical fitness tests (for non-athletes only) and psychological evaluations (survey on psychological well-being and perceived stress). We considered data only if the survey was 100% completed.

#### 2.3.2. Psychological Assessment

*Perceived Stress Scale* (PSS-10) [21,22]: This scale is a 10-item self-report measure of perceived stress during the past month. Items are rated on a 5-point Likert scale from 0 to 4, with higher scores indicating a greater level of perceived stress. The total score, which is calculated as the sum of the scores of each item, ranges between 0 (best—no perceived stress) and 40 (worst—very high perceived stress).

*World Health Organization-Five Well-Being Index* [23]: This is a self-reported short questionnaire commonly used to assess subjective well-being. It consists of five items, each focusing on different aspects of well-being. Participants were asked to rate each item on a 6-point Likert scale ranging from 0 to 5, with higher scores indicating a greater well-being level. The total score, which is calculated as the sum of the scores of each item, ranges between 0 (worst level of well-being) and 25 (better level of well-being).

*Short form of the Psychological General Well-Being Index (PGWBI a/b)* [24]: This tool evaluates both subjective well-being and psychological health. It is a brief self-report questionnaire consisting of six items, each addressing a different dimension of well-being: anxiety, depressed mood, positive well-being, self-control, general health, and vitality over the past month. Each item is rated on a 6-point Likert scale, ranging from 0 to 5, with higher scores reflecting greater subjective well-being and psychological health. Similar to the previous tools, the total score is the sum of all item scores, ranging from 0 (lowest well-being and psychological health) to 30 (highest well-being and psychological health).

#### 2.3.3. Physical Fitness Assessment

The fitness tests were administered only to the non-athlete group, as the athlete group, for obvious reasons, consisted of trained individuals who continued to follow their training routines during the study. Therefore, a home-based assessment of physical performance in the athlete group would not have provided useful information.

*2-Minute Step Test:* This test is a field test used to assess aerobic fitness. In this test, the participant must stand next to the wall and mark on the wall the corresponding point half-way between the kneecap and the iliac crest. Then, the participant, marching in place, must perform as many steps as possible, ensuring that during each step, the knee reaches the point that they previously marked on the wall, with the possibility of resting during the 2 min of the test. The final score of the test is the number of correct steps performed in 2 min.

*Curl-Up Test:* This test is a field assessment designed to evaluate the strength and endurance of the abdominal muscles. The participant lies on a mat with their knees bent at a 90° angle, feet flat on the floor, and hands resting on their thighs. The participant must then perform a series of curl-ups, sliding their hands along their thighs until the fingertips reach the top of the kneecap, before returning to the starting position. Each cycle (curl-up and return) should be completed in about 3 s. To maintain a steady pace during the test, the use of a metronome app set at 20 beats per minute is recommended. The athlete should count the number of curl-ups they can perform while keeping up with the metronome’s rhythm. The final score of the test is represented by the maximum number of correctly performed curl-ups completed until exhaustion or until the participant can no longer keep up with the metronome’s pace.

*Forward Bending Test:* This test appears to be a modified version of the traditional “Sit and Reach Test” used to assess flexibility. In particular, this provides a simple but effective way to determine overall flexibility, particularly in the hamstrings, lower back, and hip muscles. In this standing version, the participant bends forward from a standing position, attempting to reach different points on the lower limbs with their fingertips while keeping the knees extended. The breakdown of the scoring is as follows: 1 point—fingertips reach the thighs; 2 points—fingertips reach the knees; 3 points—fingertips reach the shins; 4 points—fingertips reach the ankles; 5 points—fingertips touch the feet; 6 points—fingertips touch the floor.

*Squat Test:* This field test is designed to evaluate the strength and endurance of the lower limb muscles. Participants must stand approximately 40–50 cm in front of a chair, with their back facing the chair and their feet shoulder-width apart. The participant then squats down until their buttocks lightly touch the chair (without sitting down), before returning to the standing position. This sequence of movements is repeated continuously until the participant is no longer able to continue. The total number of squats performed should be counted, with no rest allowed, and each squat cycle (down and up) should take approximately 3 s. As for the curl-up test, to maintain a steady pace during the test, it is recommended that a metronome app set at 20 beats per minute is used.

### 2.4. Statistical Analysis

The statistical analysis aimed to compare the pre- vs. post-training scores obtained from the psychological and fitness tests as well as body weight. Firstly, the Kolmogorov–Smirnov test of normality was performed on these scores to assess their normal distribution. Successively, to evaluate whether significant differences existed between the pre- and post-training scores, an analysis of variance with repeated measures (RM-ANOVA) was performed. To perform this analysis, the comparison pre-vs. post-training was considered as the time factor of the analysis (named the Time factor), while the scores obtained by the eight tests, which consisted of body weight, three psychological, and four fitness tests (these last ones for non-athletes only), were considered as the dependent variables of the analysis. To evaluate whether one or more factors could have influenced the effectiveness of the proposed training over time, the following variables were also considered as a factor of the analysis: gender of the participants (males vs. females: Gender factor), group (athletes vs. non-athletes: Group factor), mode (synchronous vs. asynchronous: Mode factor), and age group (young adult vs. adult vs. middle age vs. elderly: Age factor). Partial Eta square (η^2^*p*) was also computed as an indicator of the effect size of the analysis.

For all of the analyses, statistical significance was defined as *p* < 0.05; all results were expressed as the mean ± SD for continuous variables, or percentage and numerosity for categorical variables. All statistical analysis was performed by SPSS version 27.0 software (SPSS Inc., Chicago, IL, USA).

## 3. Results

Concerning the psychological test scores, RM-ANOVA showed a significant amelioration on well-being in the pre-post comparison assessed using the WHO-5 test (F = 16.166, η^2^*p* = 0.082, *p* < 0.001). No differences were found in the interaction Time × Gender, Time × Mode, Time × Age, and Time × Group.

Concerning the PSS-10, RM-ANOVA showed a significant increment in stress scores in the pre-post comparison (F = 59.586, η^2^*p* = 0.249, *p* < 0.001), although the mean scores still remained under the median value of the test. No differences were found in the interaction Time × Gender, Time × Age, and Time × Group, but a significant interaction Time × Mode was found (F = 18.099, η^2^*p* = 0.092, *p* < 0.001), indicating a favorable situation for asynchronous mode. However, it should be noted that the participants who trained in asynchronous mode had significantly higher scores in PSS-10 both at the baseline and after intervention.

RM-ANOVA also showed a significant amelioration of the PGWBI test scores in the pre-post comparison (F = 7.906, η^2^*p* = 0.042, *p* = 0.005). No differences were found in the interaction Time × Gender, Time × Mode, and Time × Age, but there was a significant interaction in Time × Group (F = 10.519, η^2^*p* = 0.056, *p* < 0.001), with a favorable situation for non-athletes who increased their PGBWI score over time, while the athletes slightly decreased, although they maintained scores over the median value. All of the results of the psychological tests are reported in Table 3.

Concerning the fitness tests and body weight administered to non-athletes only, RM-ANOVA did not show a significant reduction in the body weight over time (F = 2.441, η^2^*p* = 0.020, *p* = 0.121), but a significant amelioration in the pre-post comparison was found for all fitness tests: a significant amelioration was found in the 2-minute step test (F = 4.336, η^2^*p* = 0.045, *p* = 0.040), in the curl-up test (F = 6.110, η^2^*p* = 0.063, *p* = 0.015), in the forward bending test (F = 10.849, η^2^*p* = 0.107, *p* = 0.001), and in the squat test (F = 56.659, η^2^*p* = 0.384, *p* < 0.001). In all four fitness tests, only a significant interaction in Time × Gender was found in the forward bending test (F = 4.835, η^2^*p* = 0.051, *p* = 0.030) where both genders improved their flexibility over time, but the males obtained a higher increment in flexibility compared with the females. However, despite their higher increment, the males obtained an average flexibility score lower than the females both pre- and post-intervention. No other significant interactions in Time × Gender, Time × Mode, and Time × Age were seen in the fitness test.

The interaction Time × Group was not considered in the fitness test analysis because, as previously stated, the athlete group did not perform these tests. All results of the fitness tests are reported in Table 4.

In Figure 2, the main results obtained in both the psychological and physical fitness tests are reported in graphical form for more clarity.

## 4. Discussion

This study documented the effects of an 8-week fitness training program, conducted two or three times per week via tele-exercise, on target groups differing in gender, age, and physical activity habits. The main result of this study was the improvement in physical fitness across all tests among non-athletes, along with enhanced psychological well-being reported by most participants. In detail, the levels of well-being as measured through the WHO-5 and PGWBI generally increased, with the exception of the group of athletes whose scores in the PGWBI remained over the median value, with a slight decrease. The authors suggest that the athletes’ strict routine and periods of competitions may have impacted the results of this specific test that, along with the well-being factors, also explores indices of mental health such as anxiety and depressed mood. The increased levels of stress (PSS-10) may also support this hypothesis. Additionally, a careful reflection must be dedicated to the increase in stress levels also observed in the non-athletes, particularly in the synchronous mode group; the authors suggest that this could be related to the fact that the participants of both groups underwent a training program that was new to them, knowing that other strangers were attending the same lesson, and that they did not have someone physically by their side to assist and intervene in the case of problems or mistakes. Furthermore, some of the participants’ lack of familiarity with the training platform may also have played a role: the Time × Mode effect in all of the participants’ levels of stress (although median scores remained under the median values) possibly suggests that the new and technological environment of the workouts might have impacted the sense of self-efficacy of participants who exercised in a group at the virtual presence of an experienced instructor who observed them (synchronous mode), on the contrary, the lack of direct supervision in training might have resulted in a more relaxed, although not necessarily positive, attitude in the participants in asynchronous mode. Finally, it should also be noted that the general and inclusive population in the non-athlete group might have been affected by the high frequency of exercise (three times a week) compared with their sedentary habits.

The significant increase in all fitness tests revealed that the fitness programs were well-suited for all participants. Moreover, the study highlights and confirms the critical situation of young people who are becoming increasingly sedentary [25]. One effective way to counter sedentariness is through tele-exercise, which has been shown to improve both psychological well-being and physical fitness, even in individuals with disabilities. The TELEexe4ALL project demonstrated how tele-exercise can help people maintain an active lifestyle. Tele-exercise is a viable and widely recognized strategy for both improving rehabilitation and the general wellness of healthy people [14].

According to Sylvia [26], “PA is a multi-dimensional construct and thus, there is no measure that can assess all facets of PA”; key points of the TELEexe4all project are the multidisciplinary experience of the authors, the methodological approach suggested for a free platform in a scientifically regulated setting, and the use of activities that do not require any specific device.

However, despite the positive aspects, there are also some other aspects that deserve consideration. Firstly, an important aspect to highlight is that the participants included in the present study were those who completed 100% of the proposed lessons and, at the same time, completed many of the tests and surveys administered. However, it should be noted that other participants were recruited and participated in the study (see Figure 1). The largest portion of the participants that dropped out of the study participated in all of the proposed lessons but did not complete the final assessments, and therefore their data could not be used for the statistical analysis. It is possible that the pre-intervention assessments were perceived as too long and/or difficult (in particular the fitness tests), which may have led some participants to avoid completing the assessments again in the post-intervention, even though these participants were still interested in practicing tele-exercise. This information could be useful for future studies in this field, as more streamlined and immediate assessments may be more readily accepted by the participants. Another solution could be to include some tests during the initial and final lessons of the training intervention, rather than as separate moments, in order to reduce the drop-out rate during the assessments.

Secondly, as stated in the “Participants” sub-section, in the present study, a convenience sampling was used. However, this aspect should not be considered as a strict criticism: while it is true that, on the one hand, this type of sampling does not allow for complete generalization of the results, on the other hand, this method was chosen because it enabled the authors to identify the categories of people who were more inclined to participate in tele-exercise programs. In fact, the data showed that the participants in the study were primarily women and adults. This important information can be used for the development and improvement of future studies in this field of research as well as understand how to increase participation among those who are less inclined to participate in this kind of training.

It is now possible to compare the results of this study with the initial hypotheses of the authors outlined in the final part of the Introduction section:Hypothesis HP1 (tele-exercise can impact on psychological well-being levels) was substantially confirmed despite the increment in the stress levels.Hypothesis HP2 (tele-exercise can be a feasible and valuable support to set a healthy physical condition in the general population) was confirmed by the study.

## 5. Conclusions

Enabled by technology, tele-exercise provides a flexible and convenient complementary approach to conventional workouts, helping individuals maintain an active lifestyle and improve their well-being.

The TELEexe4ALL project was aimed at providing an inclusive opportunity to exercise to general sedentary populations and create an environment where individuals of different ages and from different backgrounds in PA could take care of their wellness and well-being. The project also considered the social impact of an inclusive opportunity where individuals with or without disabilities could work out together, regardless of the barriers and social stigma that often occur when people with disabilities access facilities for sport activities.

This study had some limitations. First, as previously stated, convenience sampling was selected in order to gather information on individuals who were more likely to follow this type of training, but this type of sampling did not allow us to balance the numerosity of the sub-groups according to age, gender, etc. It is likely that the heterogeneity of the sample reflects the heterogeneous levels of motivation and expectation that might have impacted on both their perception of well-being and on their levels of stress. Second, in this project, the role of social communication among participants outside the workout sessions was not specifically supported on the platform (e.g., forums, comments, groups, etc.). The inclusion of community features to allow participants to share their experiences could have played a significant role in normalizing stress levels and reducing the drop-out rates (see Appendix A). These interactions may have fostered greater socialization, provided mutual support, and enhanced motivation, all of which are probably essential for maintaining engagement in the tele-exercise program. This aspect should be considered in future projects. Third, the data relied on field tests carried out at home that was based on trusting the ability to correctly measure the performance. However, recent studies have stated that remote settings do not impact the accuracy of the results, as the difference with respect to the laboratory settings ranged from 5% to 10% [27], which is considered an acceptable error.

As a final consideration, it should be clarified that it is the authors’ opinion that tele-exercise should not be considered as an activity in contrast to in-person exercise, but rather as a complementary activity. While recognizing that each type of activity has its own specific characteristics, by complementing in-person training, tele-exercise can contribute to increasing the percentage of physically active individuals in the population, with significant benefits in terms of public health.

In the future, other objective measure techniques could be used to realize the simultaneous measurement of physical parameters (i.e., body acceleration) and/or physiological parameters (i.e., heart rate) for the assessment of physical fitness [28] and to better follow personalized improvement [29,30]; real-time monitoring could contribute to realize a safer environment for all participants. Another important aspect to consider in future research will be the implementation of platforms to make them increasingly simple and intuitive as well as the integration of artificial intelligence to monitor the accuracy and safety of the proposed exercises and other aspects related to training.

## Figures and Tables

**Figure 1 healthcare-13-00354-f001:**
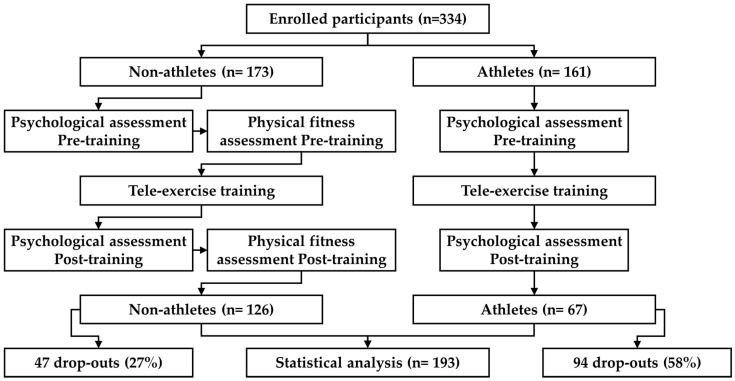
Flowchart of the study.

**Figure 2 healthcare-13-00354-f002:**
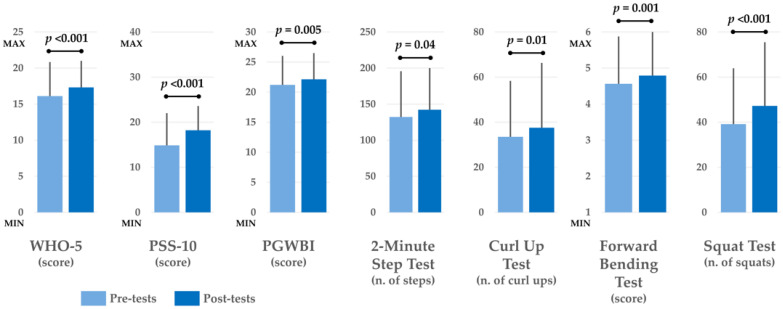
Graphical representation of the main results obtained in both the psychological and physical fitness tests. Data are displayed as means (columns) and standard deviations (error bars). The labels MIN and MAX on the *y*-axis of some tests respectively indicate the minimum and maximum values that can be obtained in those tests. Tests that do not display these labels do not have a maximum score.

**Table 1 healthcare-13-00354-t001:** Participants’ characteristics.

Group	Age (years)	Males/Females	SY/AS	Weight (kg)	Height (cm)
Athletes (67)	36.36 ± 8.95	34/33	56/11	74.81 ± 16.67	168.83 ± 9.46
Non-athletes (126)	36.79 ± 13.20	43/83	67/59	69.74 ± 14.01	169.93 ± 10.93
All (193)	36.64 ± 11.87	77/116	123/70	71.30 ± 15.00	169.67 ± 9.74

Age, weight, and height are reported as means ± standard deviation; gender and mode are reported as numerosity; AS = asynchronous; SY = synchronous.

**Table 2 healthcare-13-00354-t002:** Inclusion and exclusion criteria for enrollment.

	Inclusion Criteria	Exclusion Criteria
Athletes	• Both genders; • Athlete competing at national level (sub-elite level) and registered with a sports federation from at least 3 years; • Valid competitive medical certificate.	• Presence of pathology, disability, or other condition incompatible with the procedures proposed in this investigation; • Use of drugs or medication, or other situations that could introduce a bias into the procedures of the investigation.
Non-athletes	• Both genders; • Sedentary or normally active lifestyle; • Valid non-competitive medical certificate; • Participants of this group also included people with disabilities or other conditions compatible with the procedures proposed.	

**Table 3 healthcare-13-00354-t003:** Psychological survey results. Data are presented as the means ± standard deviation.

Factor	Participants	WHO-5 (Score)		PSS-10 (Score)		PGWBI (Score)	
		Pre	Post	Pre	Post	Pre	Post
	All (193)	16.1 ± 4.7	17.3 ± 3.7	14.9 ± 7.2	18.2 ± 5.4	21.2 ± 4.8	22.1 ± 4.3
Group	Athletes (67)	17.6 ± 4.2	18.0 ± 3.8	12.0 ± 6.4	16.6 ± 5.3	23.2 ± 4.2	22.8 ± 4.4
	Non-ath. (126)	15.3 ± 4.8	16.9 ± 3.6	16.6 ± 7.1	19.1 ± 5.2	20.0 ± 4.8	21.8 ± 4.3
Gender	Males (77)	17.5 ± 4.7	18.8 ± 3.4	12.3 ± 7.4	16.2 ± 5.9	22.8 ± 5.1	23.8 ± 4.2
	Females (116)	15.1 ± 4.5	16.3 ± 3.6	16.7 ± 6.4	19.7 ± 4.4	20.1 ± 4.3	20.9 ± 4.0
Mode	SY (123)	17.3 ± 4.4	18.1 ± 3.5	12.3 ± 7.1	17.1 ± 5.9	22.1 ± 5	22.6 ± 4.5
	AS (70)	14.3 ± 4.7	16.1 ± 3.7	19.0 ± 5.1	20.1 ± 3.7	19.7 ± 4.1	21.4 ± 3.9
Age	Young (24)	18.3 ± 4.1	19.6 ± 3.2	11.4 ± 6.8	15.3 ± 6.5	23.0 ± 5.1	24.5 ± 4.4
	Adult (142)	15.2 ± 4.7	16.6 ± 3.7	16.3 ± 6.7	19.0 ± 4.7	20.5 ± 4.6	21.6 ± 4.3
	Middle age (18)	19.2 ± 3.7	18.7 ± 3.0	9.4 ± 6.4	16.0 ± 6.4	24.0 ± 4.8	23.1 ± 4.3
	Elderly (9)	18.6 ± 2.8	19.5 ± 1.9	11.9 ± 7.9	17.9 ± 7.1	23.3 ± 4.7	23.4 ± 3.2
*p*-value	Time	*p* < 0.001		*p* < 0.001		*p* = 0.005	
	Time × Group	Not significant		Not significant		*p* < 0.001	
	Time × Gender	Not significant		Not significant		Not significant	
	Time × Mode	Not significant		*p* < 0.001		Not significant	
	Time × Age	Not significant		Not significant		Not significant	

SY = synchronous, AS = asynchronous.

**Table 4 healthcare-13-00354-t004:** Fitness tests. Data are presented as the means ± standard deviation.

Factor	Participants	2-Minute Step Test (No. of Steps)		Curl-Up Test (No. of Curl-Ups)		Forward Bending Test (Score)		Squat Test (No. of Squats)	
		Pre	Post	Pre	Post	Pre	Post	Pre	Post
	All (92)	132.2 ± 63.6	142.3 ± 57.7	33.5 ± 24.8	37.6 ± 28.7	4.6 ± 1.3	4.8 ± 1.2	39.2 ± 24.7	47.2 ± 28.3
Group	Non-ath. (92)	132.2 ± 63.6	142.3 ± 57.7	33.5 ± 24.8	37.6 ± 28.7	4.6 ± 1.3	4.8 ± 1.2	39.2 ± 24.7	47.2 ± 28.3
Gender	Males (29)	118.2 ± 49.7	138.1 ± 60.8	36.1 ± 32.5	36 ± 37.3	3.8 ± 1.2	4.2 ± 1.2	34.6 ± 19.2	40.9 ± 19.0
	Females (63)	138.6 ± 68.4	144.2 ± 56.6	32.3 ± 20.5	38.3 ± 24.2	4.9 ± 1.2	5.0 ± 1.2	41.3 ± 26.8	50.1 ± 31.4
Mode	SY (52)	124.9 ± 74.9	133.3 ± 63.4	28 ± 16.7	30.1 ± 14.5	4.3 ± 1.3	4.5 ± 1.3	35.7 ± 18.7	42.0 ± 18.6
	AS (40)	141.7 ± 43.8	154 ± 47.5	40.7 ± 31.3	47.3 ± 38.5	5.0 ± 1.2	5.2 ± 1.1	43.7 ± 30.5	54.0 ± 36.4
Age	Young (12)	96.0 ± 52.2	103.9 ± 54.2	39.5 ± 16.5	42.8 ± 13.6	4.3 ± 1.6	4.8 ± 1.6	46.9 ± 22.5	53.4 ± 24.2
	Adult (66)	139.9 ± 68.6	151.8 ± 58.7	34.9 ± 26.2	39.7 ± 31.7	4.7 ± 1.3	4.9 ± 1.2	39 ± 26.5	47.7 ± 30.9
	Middle age (6)	118.0 ± 35.5	113 ± 44.1	40.2 ± 15.8	34.3 ± 13.7	4.0 ± 0.9	4.2 ± 1.3	42 ± 19.3	46.2 ± 19.2
	Elderly (8)	133.6 ± 26.4	143.9 ± 33.9	7.8 ± 10.7	14.4 ± 16.3	4.4 ± 1.4	4.5 ± 1.2	27.3 ± 8.5	35.1 ± 10.0
*p*-value	Time	*p* = 0.040		*p* = 0.015		*p* = 0.001		*p* < 0.001	
	Time × Group	Not significant		Not significant		Not significant		Not significant	
	Time × Gender	Not significant		Not significant		*p* = 0.030		Not significant	
	Time × Mode	Not significant		Not significant		Not significant		Not significant	
	Time × Age	Not significant		Not significant		Not significant		Not significant	

SY = synchronous, AS = asynchronous.

## Data Availability

Data is available under reasonable request to the corresponding author.

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
