# Peer review of "Tele-Exercise for Fitness: Physical and Psychological Outcomes in Athletes and Non-Athletes’ Trainees"

_healthcare, 2025, doi:10.3390/healthcare13040354_

Round 1
Reviewer 1 Report
Comments and Suggestions for Authors
I would like to thank for the opportunity to review this manuscript. Please see the following comments to consider to further increase the quality of this manuscript.
The absence of a control group limits the ability to draw causal inferences. Consider adding a control group in future studies to strengthen the findings.
Convenience sampling may introduce biases. A more systematic sampling method could improve generalizability.
The unexpected increase in stress levels warrants further exploration. The manuscript could benefit from a more detailed discussion about the reasons behind this finding and potential mitigating strategies.
The imbalance in subgroup sizes (e.g., gender, age) should be acknowledged in the limitations and its potential impact on results discussed in greater depth.
Some tables (e.g., Table 4) are dense and might be better represented graphically to enhance readability.
Highlight key statistical values in bold or summarize them in an additional "Key Findings" section.
The study could include more information on participant engagement and strategies used to minimize dropout rates.
The absence of community features (e.g., forums or chats) on the TELEexe4ALL platform is mentioned as a limitation. Providing suggestions for incorporating these features could add practical value.
While the manuscript discusses the benefits of tele-exercise, it lacks a critical comparison with traditional in-person exercises. This could provide a balanced perspective.
Expand on how the findings might inform future tele-exercise program designs, especially for diverse populations or those with limited technological access.
Author Response
Please see the attached pdf file
Reviewer 2 Report
Comments and Suggestions for Authors
The main part of this manuscript is a study of the impact of teletraining on the physical fitness and mental state of people with and without disabilities.
In my opinion, the tasks that the authors set for themselves were accomplished.
As a result of using teletraining, an improvement in physical fitness and mental well-being was noted in all groups of patients.
The topic of this article is original and useful for the practical application of teletraining for all categories and all ages of people. In my opinion, this study opens up a new understanding of the possibilities of using physical exercises with the support of telesystemic training in case of limitations or problems with the availability of the sports industry.
Unlike other studies, this work provides specific recommendations for the use of appropriate teletraining programs for people with home conditions.
The results obtained correspond to the objectives of this study. In addition, the authors draw conclusions about improving the lifestyle of the sedentary population.
Please note the following two issues:
1. The dimensions of the indicators must be indicated in the tables. If these are conventional units, then this must be indicated.
2. Unfortunately, the authors did not explain the lack of reliable differences between the groups according to the psychological survey and fitness tests.
This is especially unclear in age groups.
In my opinion, all references in the article are appropriate.
I believe this article can be published.
Author Response
Please see the attached pdf file
Reviewer 3 Report
Comments and Suggestions for Authors
Abstract : Properly configured.
Introduction: The background of remote exercise and the development of remote programs in the Covid-19 situation are being carried out smoothly. However, some studies have been conducted through remote programs in the Covid-19 situation, and if additional research is presented, the need for further research will be more valid.
At the end of the introduction, please clearly present the research hypothesis you want to verify in this study
If you look at the study participants, there are elderly people over 60 years old, and if you look at the results, there are only 8 elderly people. It is questionable whether it is appropriate to check the statistics of 8 people. I propose a supplementary explanation for this part or a method of classifying it into after middle age.
In addition, we propose a plan to present height, weight, ST, and AS for each research participant group.
In the research procedure, it is said that the program was operated in various ways from 1:2 to 1:8, but there are cases where proper feedback is not received properly when a large number of people are together than a small number of people. Please explain further how you supplemented and overcame this part.
Psychological and physical evaluation are properly explained.
The data analysis method is also properly presented.
discussion
The development of the discussion is somewhat disappointing compared to the results. Rather than integrating the discussion, we propose to develop chapters such as interpretation, strengths, and limitations of the study. In addition, the discussion also plays a role in reinterpreting the results of the study, but it is necessary to mention the results, differences, and commonalities of previous studies. In the current discussion, comparative analysis with previous studies is insufficient. It seems that it has been compared and analyzed with three papers. Please develop the discussion by including more previous studies.
a general opinion
Overall, the research design was well-designed and the results were meaningful. However, it is regrettable that it seems to have been roughly written in a formal way in the discussion part. If the discussion part is reinforced, I think it will be upgraded to a better thesis.
Author Response
Please see the attached pdf file
Round 2
Reviewer 1 Report
Comments and Suggestions for Authors
Authors have done well job on revising their manuscript.